# The notable global heterogeneity in the distribution of COVID-19 cases and the association with pre-existing parasitic diseases

Taehee Chang[1], Bong-Kwang Jung[2], Jong-Yil Chai[3], Sung-il Cho[1,4]*

1 Department of Public Health Sciences, Graduate School of Public Health, Seoul National University, Seoul, Republic of Korea, 2 Institute of Parasitic Diseases, Korea Association of Health Promotion, Seoul, Republic of Korea, 3 Department of Tropical Medicine and Parasitology, Seoul National University College of Medicine, Seoul, Republic of Korea, 4 Institute of Health and Environment, Seoul National University, Seoul, Republic of Korea

* persontime@hotmail.com

**Data Availability Statement:** All epidemiological and socio-economic data are available from the Global Health Observatory (https://www.who.int/data/gho/data/indicators) and World Development

## Abstract

### Background

The coronavirus Disease 2019 (COVID-19) is a respiratory disease that has caused extensive ravages worldwide since being declared a pandemic by the World Health Organization (WHO). Unlike initially predicted by WHO, the incidence and severity of COVID-19 appeared milder in many Low-to-Middle-Income Countries (LMIC). To explain this noticeable disparity between countries, many hypotheses, including socio-demographic and geographic factors, have been put forward. This study aimed to estimate the possible association of parasitic diseases with COVID-19 as either protective agents or potential risk factors.

### Methods/Principal findings

A country-level ecological study using publicly available data of countries was conducted. We conceptualized the true number of COVID-19 infections based on a function of test positivity rate (TPR) and employed linear regression analysis to assess the association between the outcome and parasitic diseases. We considered demographic, socioeconomic, and geographic confounders previously suggested. A notable heterogeneity was observed across WHO regions. The countries in Africa (AFRO) showed the lowest rates of COVID-19 incidence, and the countries in the Americas (AMRO) presented the highest. The multivariable model results were computed using 165 countries, excluding missing values. In the models analyzed, lower COVID-19 incidence rates were consistently observed in malaria-endemic countries, even accounting for potential confounding variables, Gross Domestic Product (GDP) per capita, the population aged 65 and above, and differences in the duration of COVID-19. However, the other parasitic diseases were not significantly associated with the spread of the pandemic.

Indicators (https://databank.worldbank.org/source/world-development-indicators/Type/TABLE/preview/on#) database.

**Funding:** The author(s) received no specific funding for this work.

**Competing interests:** The authors have declared that no competing interests exist.

## Conclusions/Significance

This study suggests that malaria prevalence is an essential factor that explains variability in the observed incidence of COVID-19 cases at the national level. Potential associations of COVID-19 with schistosomiasis and soil-transmitted helminthiases (STHs) are worthy of further investigation but appeared unlikely, based on this analysis, to be critical factors of the variability in COVID-19 epidemic trends. The quality of publicly accessible data and its ecological design constrained our research, with fundamental disparities in monitoring and testing capabilities between countries. Research at the subnational or individual level should be conducted to explore hypotheses further.

## Author summary

SARS-CoV-2 has spread fast worldwide, yet the first pandemic waves in LMICs looked weaker than mathematical models had predicted. Suggestions for this observed disparity include socio-demographic and geographic factors and immunological modulation caused by exposure to endemic parasite infections. In countries where parasitic diseases are widespread, particularly in Sub-Saharan Africa (SSA), residents may have been coinfected with COVID-19 and pre-existing parasitic diseases due to the rapid spread of SARS-CoV-2. Therefore, we performed a nation-level ecological analysis to describe trends in primary SARS-CoV-2 outcomes by country and investigate potential correlations between these outcomes and pre-existing parasitic diseases. Our results suggest that malaria prevalence is a crucial factor to explain variation in COVID-19 incidence at the national scale, with a substantial relationship persisting even when putative confounders were adjusted. While we note that causal inference cannot be proved owing to insufficient data and hidden confounders, this will aid in generating new hypotheses and identifying intervention strategies to reduce NTDs and malaria in the context of the coronavirus pandemic.

## Introduction

COVID-19, caused by the novel SARS-CoV-2 virus (Coronaviridae), has reached all corners of the world, becoming the worst epidemic in the century. Since the disease outbreak, researchers have insisted on a dramatic surge of COVID-19 in Low-to-Middle-Income Countries (LMICs) compared to developed countries with better infrastructures and health policies [1,2]. However, contrary to expectations, the pandemic has caused remarkably higher morbidity and mortality consequences in developed countries. Sub-Saharan Africa (SSA), a region already beset by a lack of health infrastructure, poverty, and the highest rates of Neglected Tropical Diseases (NTDs), seem relatively spared, while other regions of the world (such as the USA and Europe) have been bearing the tremendous burden of cases and associated deaths [3]. SSA is the least affected continent, making up 2.5% of worldwide cases though accounting for 14% of the worldwide population (Fig 1).

Several hypotheses have been posited to explain why the transmission and severity of COVID-19 in LMICs remain comparatively low despite limited laboratory capacity, weak healthcare infrastructure, and a lack of public health system [2,4–10]. One argument raised is that the number of cases and deaths in the regions is underreported due to limited testing

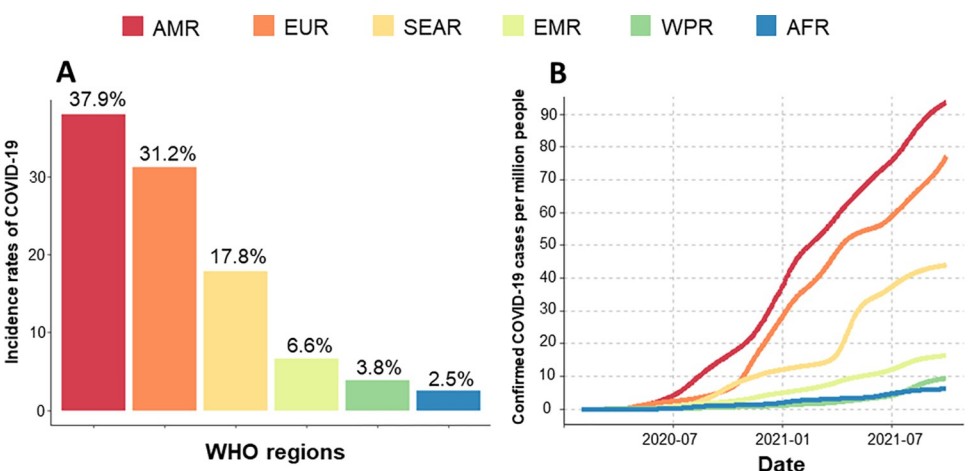

**Fig 1. Incidence rates of COVID-19 by WHO regions and their trends.** (A) The proportion of incidence rates of COVID-19 by WHO region; (B) Trends in the average number of confirmed cases per million people from March 2020 to October 2021. AMR—Americas; EUR—Europe; SEAR—Southeast Asia; EMR—East Mediterranean; WPR—Western Pacific; AFR—Africa.

capacity, though the degree is uncertain [2]. Other theories have suggested that the presence of non-communicable diseases, including diabetes, cerebrovascular diseases, and hypertension, influence the severity of COVID-19 consequences [8,10]. In LMICs, the population structure with a smaller proportion of older people who are more susceptible to infection and death could have played an essential role in curbing the spread of the coronavirus [2,5,9]. On the other hand, it has been suggested that urbanization, dense population, or cold climate can enhance coronavirus transmission, resulting in a surge of COVID-19 in developed countries [2,11]. In addition to the above factors, previous studies documented several socioeconomic indicators, including Gross Domestic Product (GDP) per capita, Human Development Index (HDI), and differences in the timing of disease onset to explain noticeable geographical differences in the burden of COVID-19 [2,5].

NTDs, including schistosomiasis and soil-transmitted helminthiases (STHs), are primarily parasitic diseases that cause significant morbidity and mortality, mainly in LMICs [12]. They affect more than 2 billion people worldwide, with disproportionately high prevalence in tropical and subtropical countries' most deprived and poorest areas [13]. Moreover, malaria is also rampant in the regions already suffering from the burden of poverty and NTDs, with more than 200 million cases per year and 400,000 associated deaths in 2018 (> 90% in SSA) [14]. With the rapid increase of COVID-19, inhabitants in the countries where the parasitic diseases are endemic, especially in SSA, have possibly been coinfected with COVID-19 and pre-existing parasitic diseases.

Human-infecting parasites have coevolved with their hosts during evolutionary processes, with persistent immune system modulation through molecular and physiological mechanisms [15–18]. Helminths or their excretory-secretory products (ES) are recognized by the host immune system, known as microorganism-associated molecular patterns (MAMPs) or pathogen-associated molecular patterns (PAMPs) [19]. Though it depends on at which stage the helminths are in their life cycle, generally, a modulated T-helper 2 (Th2) response is triggered, resulting in a hypo-responsive state, which is characterized by the stimulation of dendritic cells (DCs), eosinophils, basophils, mast cells, alternatively activated macrophages (AAMs), regulatory B cells (Bregs), and regulatory T-cells (Tregs) [17–20]. The systemic immunomodulatory effects caused by helminths are a "double-edged sword" as those can potentially support viral

**Table 1. Summary of principal characteristics of malaria, schistosomiasis, and soil-transmitted helminths.**

| Characteristics | Malaria | Schistosomiasis | Soil-transmitted helminths |
|---|---|---|---|
| **Principal symptoms** | Fever, nausea, fatigue | Acute infection causes diarrhea, abdominal pain, cough, fever, and eosinophilia; Chronic infections are often asymptomatic | Abdominal pain, nausea, anemia and diarrhea; Chronic infections are often asymptomatic |
| **Infectious agent** | *Plasmodium vivax*, *P. falciparum*, *P. malariae*, *P. ovale*, and *P. knowlesi* | *Schistosoma haematobium*, *S. japonicum*, and *S. mansoni* | *Trichuris trichiura*, *Ascaris lumbricoides*, and hookworm (*Necator americanus* and *Ancylostoma duodenale*) |

infection [21,22], affect the efficiency of vaccines [23], or protect the human host from detrimental consequences of COVID-19 by reducing inflammatory reactions [15,24,25].

Five species of the genus *Plasmodium* are known to cause malaria in humans (Table 1). Clinical symptoms of malaria occur with the rupture of infected erythrocytes, causing stimulation of immune response and resulting in the release of pro-inflammatory cytokines including IL-6, IL-12, TNF-alpha, and interferon-gamma [26]. Some patients infected with COVID-19 experience a systemic inflammatory response, called "cytokine storm," due to excessive activation of T-helper 17 (Th17) [27,28], which similarly appear in some severe manifestations of malaria induced by pro-inflammatory T-helper 1 (Th1)-related immune response [26,29]. Therefore, the resemblance of immune reactions suggests that coinfection of COVID-19 and malaria might lead to synergistic epidemics and more severe manifestations [26].

The COVID-19 outbreak has prompted researchers to conduct nationwide comparisons to assess the effect of the pandemic. However, there are particular difficulties with interpreting comparable COVID-19 data among countries with different testing capacities, political regimes, and reporting processes [30]. Models employing transmission simulations and flight data have been used to estimate the number of undetected incidents [31,32]. These methods often need location-specific inputs, which restricts their adaptability and scalability [33]. Alternatively, the infection-fatality rate (IFR) or case-fatality rate (CFR) of COVID-19 based on serological testing and extensive diagnostic testing has been utilized to determine the actual extent of the pandemic in previous cases [34,35]. However, the IFR can be impacted by variables, including the political system, demographics, and health system capability. Furthermore, the degree to which the mortality count is accurate varies depending on how COVID-19-related fatalities are defined and how testing is conducted [33]. Researchers should employ a straightforward, comprehensive measure that is scalable across all nations and regions to enable statistical methods like multiple linear regression so they may fully evaluate the real amount of morbidity brought on by COVID-19.

This study aimed to identify the potential influence of pre-existing parasitic infections on COVID-19 morbidity. The study also considered demographic, socioeconomic, and geographic variables documented in previous epidemiological studies. We conducted an ecological study using country-level data to describe patterns in COVID-19 incidence rates and explore possible associations of parasitic diseases with COVID-19 while adjusting the effect of confounding factors. Our results will be essential for developing prevention and control strategies to reduce NTDs and malaria in the context of the coronavirus pandemic.

## Methods

### Data extraction and descriptive analysis

We utilized publicly available data from each country affected by the coronavirus pandemic; observed COVID-19 confirmed cases from March 2020 to October 2021 were included [3].

**Table 2. Summary statistics of the variables included in the study.**

| Variable | n | Year | Mean | Median | sd | Min | Max |
|---|---|---|---|---|---|---|---|
| Incidence of COVID-19 cases (in %) | 202 | 2021 | 5.25 | 4.06 | 5.19 | 0 | 23.21 |
| Malaria (estimated prevalence of Malaria based on confirmed cases, in %) | 175 | 2017 | 2.58 | 0 | 7.77 | 0 | 64.5 |
| Schistosomiasis (proportion of individuals requiring preventive chemotherapy, in %) | 172 | 2019 | 4.78 | 0 | 10.17 | 0 | 50.31 |
| STHs (proportion of children requiring preventive chemotherapy, in %) | 179 | 2019 | 0.44 | 0 | 2.88 | 0 | 41.66 |
| Tuberculosis (prevalence of new and relapse cases, in %) | 187 | 2019 | 0.06 | 0.03 | 13.32 | 0 | 0.32 |
| Cause of death, by non-communicable diseases (in %) | 184 | 2019 | 70.41 | 77.48 | 20.54 | 26.99 | 96.13 |
| Diabetes (in % of population ages 20 to 79) | 202 | 2019 | 8.39 | 6.9 | 4.75 | 1 | 30.5 |
| Population ages 65 and above (in %) | 188 | 2020 | 9.23 | 7.01 | 6.53 | 1.26 | 28.39 |
| Current health expenditure (% GDP) | 187 | 2018 | 6.48 | 6.33 | 2.86 | 1.59 | 19.04 |
| GDP per capita (in $1,000) | 193 | 2018 | 16.49 | 6.61 | 24.78 | 0.27 | 114.68 |
| Human development index | 187 | 2018 | 0.72 | 0.74 | 0.15 | 0.39 | 0.97 |
| Urban population (in %) | 202 | 2020 | 61.18 | 62.11 | 22.34 | 13.35 | 100 |
| Population density (per km$^2$) | 198 | 2019 | 318.59 | 82.33 | 648.73 | 0.14 | 7915.73 |
| Average surface temperature (in ˚C) | 196 | 2017 | 20.2 | 23.77 | 8.51 | -15.41 | 30.74 |
| The number of days elapsed from the first case of COVID-19 in China until the first cases in each country (in days) | 193 | 2021 | 76.64 | 69 | 77.69 | 3 | 668 |
| Fully vaccinated with COVID-vaccine (in %) | 199 | 2021 | 37.89 | 33.24 | 26.72 | 0.04 | 87.01 |

The cumulative number of COVID-19 cases was counted and adjusted for the population size. The number of analysis units of the data is documented in Table 2.

We collated the data on predictive indicators available from various sources, including Global Health Observatory (GHO) [36] and World Development Indicators (WDI) [37]. Information on the variables employed is detailed (Table 2). Of a diverse group of human infecting parasites, only *Schistosoma* spp., STHs, and *Plasmodium* spp. were considered in this study (Table 1). The parasitic disease variables were categorized to represent the state of endemicity based on the number of confirmed cases or individuals who require preventive chemotherapy.

The epidemiological factors included in the analysis as predictive variables consist of the prevalence of tuberculosis, the proportion of death caused by non-communicable diseases, and the prevalence of diabetes. Based on the WHO report [38], these variables were selected, suggesting that patients with pre-existing medical disorders are more susceptible to coronavirus infection. Though people of all ages can be infected, older people (e.g., older than 65) are at higher risk [39]. Therefore, we considered the proportion of the population aged 65 and over as a demographic factor to reflect the age structure. As for socioeconomic variables, we used current health expenditure to measure health infrastructure, GDP per capita to measure income, and the HDI to measure overall living standard [2]. Following related literature [2,5], we considered population density, the degree of urbanization (the proportion of people residing in urban areas), and mean surface temperature (˚C) for geographic factors. We denoted several more elements, the variables that account for the timing of the disease occurrence (the number of days since the first incidence of COVID-19 in China until the first cases in each country), and the proportion of the population vaccinated with the COVID-19 vaccine.

## Outcome adjustment

We evaluated the association between the disease outcome and indicators of parasitic diseases using a linear regression model framework. However, due to uncertainties in data quality, we were only able to examine the number of reported cases as an available proxy for the real

COVID-19 outcome. Therefore, we conceptualized the true number of the infections (*Inf*) to be a function of factor *f* and reported cases *C* (Eq 1).

$$Inf = fC \tag{1}$$

To create a numerical value for *f*, two versions of *f* ($f_1$ and $f_2$) were considered following a previous study [33]; $f_1$ was obtained using CFR, while $f_2$ utilized TPR. The factor $f_1$ was suggested to effectively reflect *Inf* since CFR has been employed as a real-time way to monitor underreporting and an indication of the actual prevalence of an infectious illness in previous situations [34,35]. Nonetheless, mortality statistics may also be underreported in areas with insufficient or low testing levels, making CFR a less representative indication. Therefore, in this study, the simpler and more comprehensive factor $f_2$ was utilized to estimate *Inf* (Eq 2), based on test positivity rate (TPR)—an indication of appropriate testing in relation to disease prevalence (Eq 3). The data on the number of tests per million was collected from WDI [37].

$$f_2 = \begin{cases} 1, & if\ TPR < 0.05 \\ TPR/0.05, & if\ TPR \geq 0.05 \end{cases} \tag{2}$$

$$TPR = \frac{C}{T} \tag{3}$$

When TPR is less than 5%, the World Health Organization (WHO) recommends that testing capacity is acceptable [40]. If TPR was more than 5 percent, it was assumed that a growing number of instances were being unreported and that the factor $f_2$ would be equal to the TPR's ratio to that of 5 percent, the threshold (Eq2). The estimated *Inf* was used as a response variable in linear regression models in this study.

## Statistical modeling

The β coefficients and 95% confidence intervals (95% CIs) were estimated based on the root-transformed COVID-19 incidence rates. The level of statistical significance was set at 0.05. All analysis was conducted on R software [41]. As the response variable was continuous, a least-squares fixed-effect linear regression model was applied to explore possible associations.

We considered the square root transformation of the outcome variable, *Inf*, since the variable was not normally distributed. Then, we investigated whether continuous predictive variables could be assumed to have linearity with root-transformed COVID-19 incidence rates by fitting a generalized additive model (GAM). Most predictors had a non-linear relationship with the response variable and showed better fitting once categorized than continuous forms. Therefore, to maintain consistency through the variables, all predictive variables were categorized into ordered quantiles or according to clusters that appeared on a scatterplot.

Classic standard errors (SE) may not be accurately estimated due to the small sample size and skewed outcome distribution. We, therefore, presented both traditional SE and 95% CI, and robust SE and 95% CI computed using the bootstrap method (boot package, R = 2,000) [42,43] as a complementary estimation approach.

## Multivariable linear regression

We applied univariable and multivariable linear regression analysis to investigate the associations. All predictive variables were tested first in univariable regression, and only the variables significantly associated with the outcome were included in the multivariable model. To avoid the problem of multicollinearity, we first observed pair-wise Pearson correlation coefficients

and scatterplots (S1 Fig). We screened variables that could cause multicollinearity, and within a group of correlated variables, the variables with fewer missing values and more substantial statistical power were selected. Subsequently, a stepwise forward selection procedure based on parasitic disease variables retained the number of predictors to the level at which the adjusted $R^2$ ($R_a^2$) is not higher above the $R_a^2$ of the complete model. The selection step was carried out by referring to the study that documented the importance of restraining alpha significance level and preventing overestimation of the explained variance [44]. We fit the models with $R_a^2$ and F-statistic, testing whether the models fit data better than the null models, and verified that the final model presents the least value of Akaike Information Criterion (AIC) and Bayesian Information Criterion (BIC). We also confirmed model assumptions, including the homoscedasticity and normality of the model residuals. To further investigate the causal framework rather than the simple correlation between the response variable and the parasitic diseases, the prevalence of parasitic diseases, which showed significant results in the final model, was analyzed as a continuous variable using univariable regression.

## Results

### Observed patterns through countries

The cumulative number of COVID-19 confirmed cases from March 2020 to October 2021 was 246,594,191, while the associated deaths have risen to 4,998,784 worldwide [3]. Among 202 countries included in the data, the countries in the Americas (AMR) showed the highest rates of COVID-19 incidence, followed by Europe (EUR), Southeast Asia (SEAR), East Mediterranean (EMR), Western Pacific (WPR), and Africa (AFR) (Fig 1). This trend has been consistent since the onset of the disease. The distribution of parasitic diseases was also heterogeneous across regions (Fig 2). The countries endemic for malaria (21.2%, 42/175), schistosomiasis (22.3%, 45/172), or STHs (26.3%, 52/179) are distributed mainly in AFR and WPR, the regions least affected by COVID-19.

### Linear regression models

A crude univariable regression analysis revealed characteristics among countries and predictive variables that were significantly associated with the risk of COVID-19 infection (Table 3). The countries with prevalent parasitic diseases presented relatively lower COVID-19 incidence rates. In the case of confounding factors, countries with a higher prevalence of tuberculosis, higher surface temperature, and later onset of COVID-19 also showed lower COVID-19 incidence rates. In contrast, the proportion of the population over 65, GDP per capita, current health expenditure per GDP, the degree of urbanization, the proportion of death by non-communicable diseases, Human Development Index, mean years of schooling, and the percentage of the population fully vaccinated with COVID-19 vaccine were positively associated with the incidence rates of COVID-19.

A multivariable model with parasitic diseases including malaria, schistosomiasis, and STHs was analyzed as a fundamental model (Table 4). COVID-19 incidence rates were significantly lower in countries endemic for malaria (β coefficient = -1.38; 95% CI of -2.33 to -0.41; $p < 0.01$). The countries where schistosomiasis or STHs are prevailing, on the other hand, were not revealed to have a significant association with COVID-19 incidence rates, though the coefficients were both negative for schistosomiasis (β coefficient = -0.84; 95% CI of -1.76 to 0.15; $p = 0.09$) and STHs (β coefficient = -0.58; 95% CI of -1.20 to 0.01; $p = 0.05$).

After screening significant variables using univariable analysis, variables with pairwise correlation coefficients between -0.7 and 0.7 were comprised for further multivariable linear

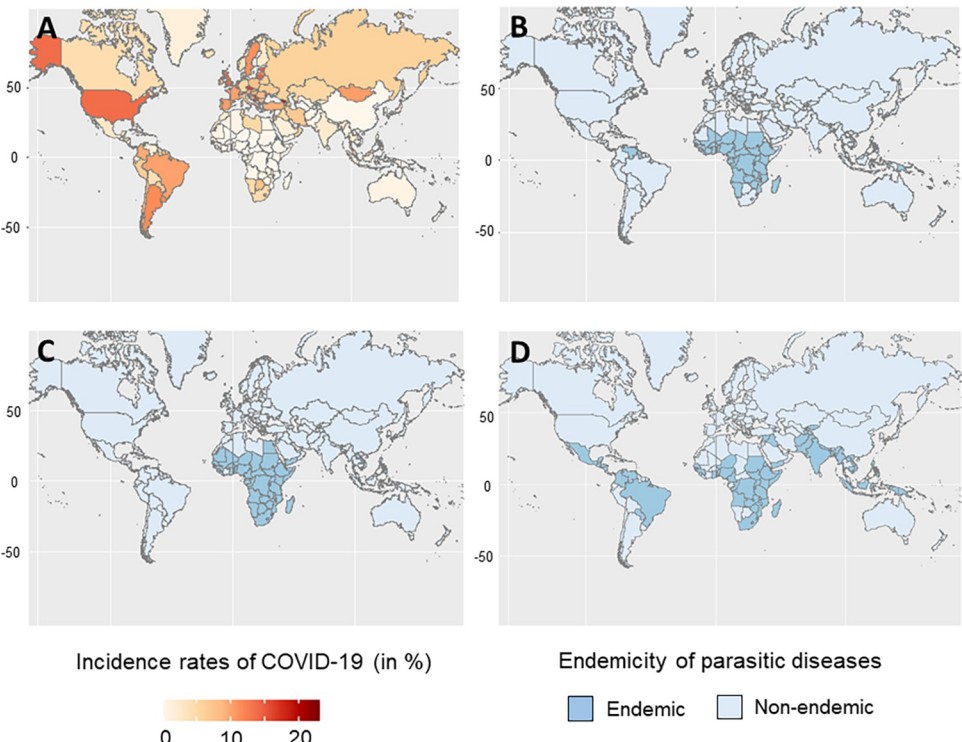

**Fig 2. Differential distribution of COVID-19 and parasitic diseases.** (A) COVID-19; (B) malaria; (C) schistosomiasis; (D) soil-transmitted helminthiasis. The map data named "world" in the R software package "maps" was employed to draw base maps in the figure. (https://rdocumentation.org/packages/maps/versions/3.4.0).

regression (S1 Fig). The saturated model had the least value of AIC and BIC (S1 Table). Since the AIC and BIC significantly increased when the parasitic disease variables were removed from the model, it could be concluded that the variables explain a considerable level of variation in the saturated model. A total of 165 countries were retained in the model, excluding missing values. Table 5 summarizes key results from the multivariable regression model of imputed predictive variables for the response variable, COVID-19 incidence rates. In the model, the malaria-endemic countries presented significantly lower COVID-19 incidence rates (β coefficient = -0.73; 95% CI of -1.46 to -0.10; $p < 0.05$) even adjusting for confounding variables, the timing of the disease eruption (duration variable), the proportion of the population over 65, and GDP per capita, while schistosomiasis (β coefficient = -0.30; 95% CI of -1.02 to 0.40; $p = 0.39$) and STHs (β coefficient = -0.14; 95% CI of -0.67 to 0.39; $p = 0.59$) were not significantly associated with the response variable. In addition, the model examined GDP per capita to have significant associations with COVID-19 incidence rates. In the univariable regression model with the prevalence of malaria as a predictor, and COVID-19 incidence rates as an outcome, the outcome variable monotonically decreased as the predictor increased (β coefficient = -0.02; 95% CI of -0.04 to -0.01; $p < 0.01$) (Fig 3).

## Discussion

We performed an ecological study to reveal crucial factors that may explain the worldwide disparity of SARS-CoV-2 epidemics, focusing on the significance of exposure to endemic parasitic infections. We identified that malaria prevalence is an essential factor explaining variation in observed case incidence of COVID-19 at the national level, with a significant association

**Table 3. Univariable analyses of predictive variables and root-transformed COVID-19 incidence rates.**

| Variable | Number of countries examined (%)[1] | Coefficient (95% CI) | SE of coefficient | Coefficient (95% CI) /bootstrapping | SE of coefficient /bootstrapping | p-value of coefficient[2] | ΔR² |
|---|---|---|---|---|---|---|---|
| **Malaria (in %)** | | | | | | | |
| Non-endemic | 156 (78.79) | Reference | | Reference | | | |
| Endemic | 42 (21.21) | -2.29 (-2.89, -1.69) | 0.3 | -2.29 (-2.64, -1.86) | 0.19 | < 0.001 | 0.23 |
| **Schistosomiasis (in %)** | | | | | | | |
| Non-endemic | 157 (77.73) | Reference | | Reference | | | |
| Endemic | 45 (22.27) | -2.19 (-2.82, -1.56) | 0.31 | -2.19 (-2.63, -1.75) | 0.3 | < 0.001 | 0.19 |
| **STHs (in %)** | | | | | | | |
| Non-endemic | 146 (73.74) | Reference | | Reference | | | |
| Endemic | 52 (26.26) | -1.49 (-2.13, -0.85) | 0.32 | -1.49 (-2.06, -0.86) | 0.3 | < 0.001 | 0.11 |
| **Tuberculosis (in %)** | | | | | | | |
| Low (< 0.01) | 55 (29.41) | Reference | | Reference | | | |
| Moderate (< 0.1) | 91 (48.66) | -0.99 (-1.63, -0.35) | 0.32 | -0.99 (-1.77, -0.61) | 0.36 | < 0.001 | 0.17 |
| High (0.1 ≤) | 41 (21.93) | -2.38 (-3.16, -1.61) | 0.39 | -2.38 (-3.11, -1.72) | 0.34 | < 0.001 | |
| **Population ages 65 and above (in %)** | | | | | | | |
| < 5 | 72 (38.30) | Reference | | Reference | | | |
| < 15 | 70 (37.23) | 2.15 (1.54, 2.75) | 0.35 | 2.15 (1.54, 2.80) | 0.34 | < 0.001 | 0.26 |
| 15 ≤ | 46 (24.47) | 2.28 (1.60, 2.97) | 0.32 | 2.28 (1.74, 2.88) | 0.28 | < 0.001 | |
| **Death by non-communicable diseases (in %)** | | | | | | | |
| < 60 | 50 (27.17) | Reference | | Reference | | | |
| < 80 | 54 (29.35) | 1.96 (1.26, 2.65) | 0.35 | 1.96 (1.32, 2.70) | 0.17 | < 0.001 | 0.25 |
| 80 ≤ | 80 (43.48) | 2.51 (1.87, 3.15) | 0.32 | 2.51 (2.03, 2.96) | 0.14 | < 0.001 | |
| **Diabetes (in %)** | | | | | | | |
| < 5 | 39 (19.31) | Reference | | Reference | | | |
| < 10 | 103 (50.99) | 1.17 (0.41, 1.93) | 0.38 | 1.17 (0.53, 1.78) | 0.31 | < 0.01 | 0.05 |
| 10 ≤ | 60 (29.70) | 1.13 (0.29, 1.96) | 0.41 | 1.13 (0.38, 1.98) | 0.4 | < 0.01 | |
| **GDP per capita (in $)** | | | | | | | |
| < 2000 | 48 (24.87) | Reference | | Reference | | | |
| < 6,000 | 43 (22.28) | 1.70 (0.99, 2.43) | 0.36 | 1.70 (1.13, 2.36) | 0.3 | < 0.001 | 0.31 |
| < 18,000 | 53 (27.46) | 3.02 (2.33, 3.71) | 0.34 | 3.02 (2.41, 3.73) | 0.33 | < 0.001 | |
| 18,000 ≤ | 49 (25.39) | 2.26 (1.56, 2.96) | 0.35 | 2.26 (1.86, 2.69) | 0.21 | < 0.001 | |
| **Current health expenditure (% GDP)** | | | | | | | |
| < 5 | 60 (38.08) | Reference | | Reference | | | |
| < 8 | 81 (43.32) | 1.59 (0.94, 2.24) | 0.32 | 1.59 (0.94, 2.24) | 0.27 | < 0.001 | 0.14 |
| 8 ≤ | 46 (24.60) | 1.74 (1.01, 2.49) | 0.37 | 1.74 (1.01, 2.49) | 0.35 | < 0.001 | |
| **Average surface temperature (in ˚C)** | | | | | | | |
| Polar-Microthermal (< 18) | 66 (33.67) | Reference | | Reference | | | |
| Temperate (< 26) | 60 (30.61) | -1.25 (-1.93, -0.51) | 0.36 | -1.25 (-1.85, -0.55) | 0.33 | < 0.001 | 0.11 |

(*Continued*)

**Table 3.** (Continued)

| Variable | Number of countries examined (%)[1] | Coefficient (95% CI) | SE of coefficient | Coefficient (95% CI) /bootstrapping | SE of coefficient /bootstrapping | p-value of coefficient[2] | ΔR² |
|---|---|---|---|---|---|---|---|
| Tropical ($\leq 26$) | 70 (35.71) | -1.27 (-1.96, -0.58) | 0.34 | -1.27 (-1.88, -0.44) | 0.36 | < 0.001 | |
| **Urban population (in %)** | | | | | | | |
| < 50 | 67 (33.17) | Reference | | Reference | | | |
| < 75 | 72 (35.64) | 1.91 (1.27, 2.56) | 0.32 | 1.91 (1.33, 2.60) | 0.32 | < 0.001 | 0.17 |
| 75 $\leq$ | 63 (31.19) | 1.69 (1.02, 2.35) | 0.33 | 1.69 (1.15, 2.24) | 0.27 | < 0.001 | |
| **Fully vaccinated with COVID-vaccine (in %)** | | | | | | | |
| < 30 | 86 (43.21) | Reference | | Reference | | | |
| < 60 | 61 (30.65) | 2.20 (1.58, 2.82) | 0.31 | 2.20 (1.55, 2.82) | 0.32 | < 0.001 | 0.3 |
| 60 $\leq$ | 52 (26.13) | 1.27 (0.67, 1.92) | 0.33 | 1.27 (0.71, 2.11) | 0.33 | < 0.001 | |
| **Population density (per km²)** | | | | | | | |
| < 50 | 67 (33.83) | Reference | | Reference | | | |
| < 120 | 59 (29.80) | 0.36 (-0.37, 1.65) | 0.37 | 0.36 (-0.36, 1.45) | 0.34 | 0.34 | 0.02 |
| 120 $\leq$ | 72 (36.36) | 0.20 (-0.50, 0.98) | 0.35 | 0.20 (-0.47, 0.88) | 0.36 | 0.57 | |
| **Human development index** | | | | | | | |
| < 0.6 | 46 (24.60) | Reference | | Reference | | | |
| < 0.8 | 76 (40.64) | 2.43 (1.80, 3.06) | 0.32 | 2.43 (1.93, 2.97) | 0.32 | < 0.001 | 0.31 |
| 0.8 $\leq$ | 65 (34.76) | 2.85 (2.20, 3.51) | 0.33 | 2.85 (2.48, 3.33) | 0.33 | < 0.001 | |
| **Duration (in days)** | | | | | | | |
| < 70 | 104 (52.26) | Reference | | Reference | | | |
| 70 $\leq$ | 95 (47.74) | -1.34 (-1.89, -0.79) | 0.28 | -1.34 (-1.89, -0.78) | 0.27 | < 0.001 | 0.19 |
| **Mean years of schooling (in years)** | | | | | | | |
| < 7 | 58 (31.35) | Reference | | Reference | | | |
| < 10 | 50 (27.03) | 1.59 (1.04, 1.77) | 0.33 | 1.59 (1.11, 1.67) | 0.28 | < 0.001 | 0.3 |
| 10 $\leq$ | 77 (41.62) | 1.97 (1.54, 2.20) | 0.29 | 1.97 (1.64, 2.19) | 0.31 | < 0.001 | |

[1]Percentages reflect cases without missing values

[2]The level of statistical significance is 0.05

retained even when possible confounders were adjusted. Potential associations of schistosomiasis and STHs with SARS-CoV-2 infection appeared unlikely, from this study, to be critical factors in the heterogeneity in SARS-CoV-2 epidemic trends. Overall, while we emphasize that causal inference cannot be concluded due to incomplete data and hidden confounders, we discovered that the prevalence of malaria has a consistent association that indicates its impact on COVID-19 epidemics, indicating that further research is needed.

The malaria prevalence was the only measure of parasitic infection that consistently showed a strong negative association with COVID-19 incidence rates. One plausible biological mechanism might be the variable distribution of the polymorphisms associated with angiotensin-converting enzyme 2 (ACE2). A type I transmembrane amino-peptidase, ACE2, is anchored at the surface of cells of the heart, kidneys, gastrointestinal system, blood vessels, and type II alveolar cells of the lungs [45]. Angiotensin II (ANG II), the substrate for ACE2, was reported to impair *Plasmodium* development, directly disturbing the protozoa membrane and protecting

**Table 4. Summary of associations between parasitic disease variables and root-transformed COVID-19 incidence rates.**

| Variable[1] | Coefficient (95% CI) | SE of coefficient | Coefficient (95% CI) /bootstrapping | SE of coefficient /bootstrapping | p-value of coefficient | Adjusted ΔR² | p-value of ΔR² |
|---|---|---|---|---|---|---|---|
| **Malaria (in %)** | | | | | | | |
| Non-endemic | Reference | | Reference | | | | |
| Endemic | -1.38 (-2.33, -0.41) | 0.48 | -1.38 (-2.06, -0.63) | 0.36 | < 0.01 | | |
| **Schistosomiasis (in %)** | | | | | | | |
| Non-endemic | Reference | | Reference | | | 0.24 | < 0.001 |
| Endemic | -0.84 (-1.76, 0.15) | 0.48 | -0.84 (-1.57, -0.04) | 0.38 | 0.09 | | |
| **STHs (in %)** | | | | | | | |
| Non-endemic | Reference | | Reference | | | | |
| Endemic | -0.58 (-1.20, 0.01) | 0.32 | -0.58 (-1.15, 0.02) | 0.32 | 0.05 | | |

[1]Interaction terms were not significant

the human host from malaria [46,47]. The presence of genetic deletion/insertion (D/I) and C1173T substitution polymorphisms are linked to changes in the concentration of ACE2, reducing ACE2 expression in the dominance of the D allele and T allele, respectively [46].

**Table 5. Summary of multivariable linear regression, including parasite variables of interest and potential confounders.**

| Variable | Coefficient (95% CI) | SE of coefficient | Coefficient (95% CI) /bootstrapping | SE of coefficient /bootstrapping | p-value of coefficient | Adjusted ΔR² | p-value of ΔR² |
|---|---|---|---|---|---|---|---|
| **Malaria** | | | | | | | |
| Non-endemic | Reference | | Reference | | | | |
| Endemic | -0.73 (-1.46, -0.10) | 0.36 | -0.73 (-1.42, -0.12) | 0.34 | < 0.05 | | |
| **Schistosomiasis** | | | | | | | |
| Non-endemic | Reference | | Reference | | | | |
| Endemic | -0.30 (-1.02, 0.40) | 0.36 | -0.30 (-1.06, 0.34) | 0.36 | 0.39 | | |
| **STHs** | | | | | | | |
| Non-endemic | Reference | | Reference | | | | |
| Endemic | -0.14 (-0.67, 0.39) | 0.26 | -0.14 (-0.64, 0.45) | 0.28 | 0.59 | | |
| **Duration (in days)** | | | | | | | |
| < 70 | Reference | | Reference | | | 0.49 | < 0.001 |
| 70 ≤ | -0.40 (-0.87, 0.07) | 0.24 | -0.40 (-0.88, 0.09) | 0.29 | 0.11 | | |
| **Population ages 65 and above (in %)** | | | | | | | |
| < 5 | Reference | | Reference | | | | |
| < 15 | 0.17(-0.41, 0.74) | 0.29 | 0.17(-0.45, 0.80) | 0.33 | 0.56 | | |
| 15 ≤ | 0.35 (-0.38, 1.04) | 0.35 | 0.35 (-0.35, 1.02) | 0.36 | 0.36 | | |
| **GDP per capita (in $)** | | | | | | | |
| < 2000 | Reference | | Reference | | | | |
| < 6,000 | 0.88 (0.25, 1.51) | 0.31 | 0.88 (0.26, 1.58) | 0.32 | < 0.001 | | |
| < 18,000 | 1.86 (1.12, 2.59) | 0.37 | 1.86 (1.26, 2.64) | 0.34 | < 0.001 | | |
| 18,000 ≤ | 0.96 (0.12, 1.80) | 0.41 | 0.96 (0.22, 1.78) | 0.39 | < 0.05 | | |

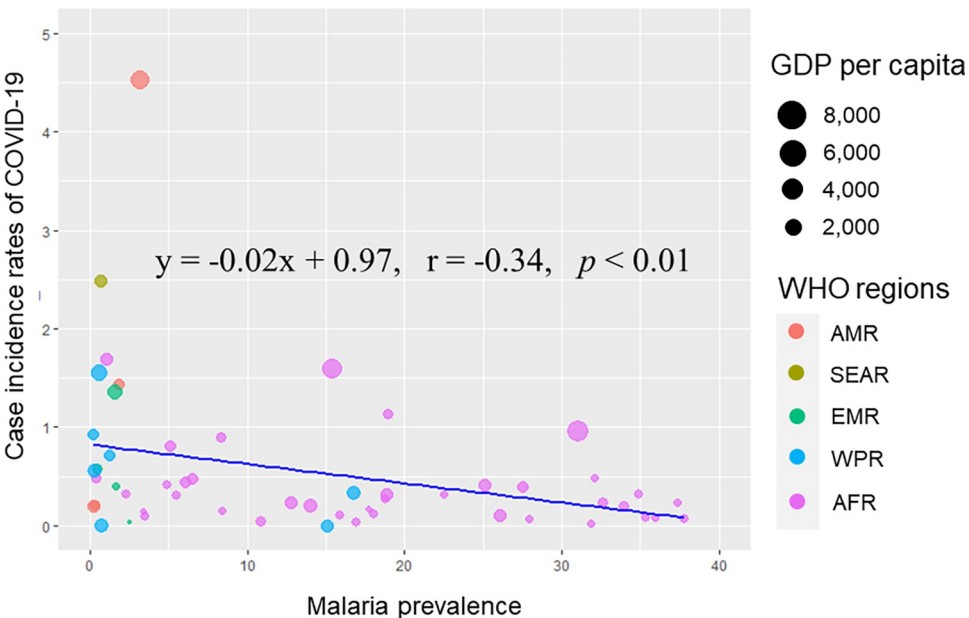

**Fig 3. Scatter plot diagram of the malaria prevalence and case incidence rates of COVID-19 for malaria-endemic countries.**

Subsequently, ANG II plasma levels are increased due to the reduction of ACE2, alleviating the clinical course of malaria, which may be a powerful driver of natural selection in malaria-endemic regions. SARS-CoV-2 requires the ACE2 receptor to enter the host cells, especially the type II alveolar cells [48]. The downregulation of ACE2 receptors due to the polymorphisms may result in a "protective" effect influencing the cell entry of SARS-CoV-2 [49]. Therefore, this mutual influence may explain the differential distribution of COVID-19 among countries, though additional genetic studies are needed to better understand the protective role of ACE2 polymorphisms.

Another hypothesis proposed from the literature was about the cross-reactivity of interferons and the neutralizing antibodies. The CD-147 receptor, expressed on several blood cells and immune cells, is a common entryway for both coronavirus and malarial *Plasmodium* to invade host cells [50]. Reports from previous studies have shown that lymphocytes produced interferons as an immune response to infection by some strains of *Plasmodium*; since the pathogens share the receptor for invasion, these interferons have a protective effect against the coronaviruses responsible for MERS, SARS, and COVID-19 [51–53]. The contribution of CD-147 blockers, Azithromycin, to preventing coronavirus infection also serves as evidence for the hypothesis [50]. Moreover, prior exposure to malaria enhances the production of *Plasmodium*-specific IgG antibodies targeting glycosylphosphatidylinositol (GPI) molecules, also capable of interacting with various glycoproteins (GPs) of SARS-CoV-2, which may confer some resistance against the coronavirus infection [54]. Although previous malaria infections cannot fully protect individuals from the disease, the severity of clinical presentation in individuals who repeatedly encountered malaria infections is milder than in "non-immune" subjects [55], suggesting that this scenario may occur in the cases of coronavirus infection.

The problems in early diagnosis due to the similar clinical features between malaria and COVID-19 may also have caused the lower number of reported COVID-19 cases in malaria-endemic regions. The first step to identifying a COVID-19 case is symptomatic screening. Early symptoms of SARS-CoV-2 infection such as fever, difficulty in breathing, and fatigue

could be confused with symptoms of malaria, leading to the misdiagnosis of COVID-19 patients as malaria and vice versa [46,49]. Patients with clinical symptoms may be tested for *Plasmodium* infection when they actually have COVID-19 and then ignored due to a negative result. Therefore, especially in malaria-endemic areas, it would be recommended to double-screen patients with suspected symptoms of both malaria and COVID-19.

Helminth infection has various immunomodulatory effects, leading to enhanced resistance to some pathogens, increased susceptibility to others, and variations in the severity of autoimmune, allergic, and inflammatory diseases [23,56]. Moreover, several studies proposed immunomodulatory properties of helminths, which may result in enhanced antiviral mechanisms, such as the role of *Fasciola hepatica* products in the host [16], a relatively milder course of COVID-19 in *Wuchereria bancrofti* and SARS-CoV-2 co-infection [24], lower risk of respiratory viral infections in mice with schistosomiasis [57], limited inflammatory damage induced by influenza virus in *Trichinella spiralis* infected hosts [58], and indirect protective strategy through changes of diversity and composition of the microbiome [15]. It has also been suggested that helminth infection may account for better tolerance of viral infections, including SARS-CoV-2 [23,59]. However, some helminths may exacerbate the clinical course of viral infection. As Siles-Lucas et al. [16] discussed, *Heligmosomoides polygyrus* and *T. spiralis* infection in mice could enhance viral infections; *S. mansoni* induces a pro-inflammatory response in the early stages of the infection [60], which may lead to synergistic epidemics and more severe manifestations of COVID-19. Overall, potential associations of schistosomiasis and STHs with COVID-19 appeared unlikely, at least based on this study, to affect the differential distribution of COVID-19. Analysis with aggregated data may be insufficiently granular to discover a potential influence of endemic infections. A previous study [61] reported that within-country variation in an immunological profile associated with changes in exposure to diseases could be detected. Our analysis at the national level could not fully reflect this heterogeneity.

Our findings present that GDP per capita, the measure of income, is significantly associated with the variable distribution of COVID-19. According to a large body of research, this factor leads to better health outcomes by allowing for improved health infrastructure and the funding of effective containment strategies such as social distancing [2,62,63]. However, it warrants further research to determine why "richer" countries with the capacity to control the spread of the disease effectively have higher incidence rates of COVID-19 in the multivariable model. As described in a previous study [2], the variable that presents the duration of the pandemic in each country adjusted the effect of disparities in the timing of disease onset, confirming whether the difference in COVID-19 distribution between countries is simply because each country is in the different phases of the pandemic. On the other hand, the proportion of the population aged 65 and above was not significantly associated with the transmission of COVID-19, based on this study. The factor was suggested as an essential demographic indicator capturing population aging and affecting the morbidity of COVID-19 in previous literature [2,9,39], which contrasts with the results of this study. Although this result may be due to the limitation of ecological research and incomplete data, one possible hypothesis is the presence of endogenous behavioral reactions. Since they are more vulnerable, older people are more likely to be cautious in implementing social distancing measures and safety precautions than others, which may finally reduce their greater infection risk, resulting in no statistical significance in an empirical analysis.

In April 2020, because of the COVID-19 pandemic, WHO declared a general recommendation to disrupt intervention programs for Neglected Tropical Diseases (NTDs). The impact of this interruption, paired with COVID-19, will reverberate for a year. The complicated effects of the COVID-19 pandemic might result in an additional 200 million malaria infections and 380,000 associated deaths in sub-Saharan Africa [64]. Similarly, it is predicted to require extra

time to cope with increased infection levels of two other infectious diseases, schistosomiasis and STHs, whose prevention programs were interrupted by the efforts to reduce the COVID-19 spread [25]. The current global priority of protecting people from attaining the SARS-CoV-2 infection, resulting in the interruption of activities of the other health sectors, makes the already vulnerable part of the population even more fragile and "neglected."

Despite the strengths of this study, there are several limitations to be noted. Related to the ecological design, the results of this study might be biased due to misclassification of the exposure to endemic diseases. Although within-country variation is frequently observed in the distribution of parasitic diseases, including malaria, schistosomiasis, and STHs, the aggregated nature of the data could not fully reflect this heterogeneity. Therefore, individual-level data with precise locations are needed to assess the exact correlation between the coronavirus and indigenous diseases. The other significant limitation is the misclassification of COVID-19 cases due to testing capacity and surveillance quality, likely to vary among countries. Due to the differences in health infrastructure and testing policy, the data may obscure the accurate size of the diseases [5]. Studies have revealed significantly under-reported cases and associated deaths in several countries, especially in low-income countries [65]. Even though we conceptualized the true number of the infections based on TPR to adjust possible biases in the under-reporting of the COVID-19 cases, we cannot conclude that our study has perfectly precluded them. Thus, our results should be cautiously interpreted. We also want to point out that we might not have adjusted for all confounding variables, though we attempted to include demographic, socioeconomic, and geographic variables suggested in previous studies. Our study is based on an analysis describing a still evolving and imperfectly explored epidemic, only considering the scientific understanding of the disease at the time. Therefore, due to these unknown factors, our study could not deal with all factors that influence the spread of COVID-19 and residual confounding.

This study aimed to assess the role of parasitic diseases in explaining the spread of COVID-19 and to understand why the disease is progressing at a slower pace in LMICs. Malaria prevalence appears to be an essential element associated with the epidemics of SARS-CoV-2, even after potential confounding factors were adjusted. Our analyses suggest significantly lower COVID-19 incidence rates in malaria-endemic regions, showing a negative dose-response relationship. However, our findings do not conclusively support the causal inference of endemic diseases to SARS-CoV-2 transmission. Although we obtained results indicating that malaria infection has a kind of "protective effect" against coronavirus, further investigation at the subnational or individual level should be conducted on the interaction of SARS-CoV-2 with various human infecting parasites to prevent unexpected harm. The COVID-19 pandemic will have long-term social, economic, and health consequences worldwide. The researchers are responsible for advocating for the resume of intervention programs for malaria and NTDs on the global health agenda. When the world is prioritizing the historical pandemic, this moment could be pivotal to strengthening the health systems that prevent malaria and NTDs.

## Supporting information

**S1 Fig. Variable collinearity heatmap.** Values in each cell are the Pearson correlation coefficient for a given pair of variables. Coefficients close to -1 or +1 indicate high correlation, with 0 indicating minimal correlation.
(TIF)

**S1 Table. AIC and BIC of hierarchical models.**
(PDF)

## Author Contributions

**Conceptualization:** Bong-Kwang Jung, Jong-Yil Chai, Sung-il Cho.

**Data curation:** Taehee Chang.

**Formal analysis:** Taehee Chang.

**Investigation:** Taehee Chang.

**Methodology:** Taehee Chang.

**Software:** Taehee Chang.

**Supervision:** Sung-il Cho.

**Validation:** Bong-Kwang Jung, Jong-Yil Chai, Sung-il Cho.

**Visualization:** Taehee Chang.

**Writing – original draft:** Taehee Chang.

**Writing – review & editing:** Taehee Chang, Bong-Kwang Jung, Jong-Yil Chai, Sung-il Cho.

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
