## [Decision Letter · Decision Letter 0]

16 Jun 2022

Dear Prof. Cho,

Thank you very much for submitting your manuscript "The notable global heterogeneity in the distribution of COVID-19 cases and the association with pre-existing parasitic diseases" for consideration at PLOS Neglected Tropical Diseases. As with all papers reviewed by the journal, your manuscript was reviewed by members of the editorial board and by several independent reviewers. In light of the reviews (below this email), we would like to invite the resubmission of a significantly-revised version that takes into account the reviewers' comments. 

We cannot make any decision about publication until we have seen the revised manuscript and your response to the reviewers' comments. Your revised manuscript is also likely to be sent to reviewers for further evaluation.

Sincerely,

Alberto Novaes Ramos Jr

Associate Editor

Liesl Zuhlke

Deputy Editor

Reviewer's Responses to Questions

**Key Review Criteria Required for Acceptance?**

**Methods**

-Are the objectives of the study clearly articulated with a clear testable hypothesis stated?

-Is the study design appropriate to address the stated objectives?

-Is the population clearly described and appropriate for the hypothesis being tested?

-Is the sample size sufficient to ensure adequate power to address the hypothesis being tested?

-Were correct statistical analysis used to support conclusions?

-Are there concerns about ethical or regulatory requirements being met?

Reviewer #1: Yes.

Reviewer #2: The study was performed with appropriate methods.

Reviewer #3: (No Response)

**Results**

-Does the analysis presented match the analysis plan?

-Are the results clearly and completely presented?

-Are the figures (Tables, Images) of sufficient quality for clarity?

Reviewer #1: Yes.

Reviewer #2: Yes.

Reviewer #3: (No Response)

**Conclusions**

-Are the conclusions supported by the data presented?

-Are the limitations of analysis clearly described?

-Do the authors discuss how these data can be helpful to advance our understanding of the topic under study?

-Is public health relevance addressed?

Reviewer #1: Yes.

Reviewer #2: Yes.

Reviewer #3: (No Response)

**Editorial and Data Presentation Modifications?**

Reviewer #1: "Minor revision"

Reviewer #2: (No Response)

Reviewer #3: (No Response)

**Summary and General Comments**

Reviewer #1: The aim of the study, its construction and the problem it addressed were quite well analysed. The relationship between COVID-19, which played an active role in the last 2.5 years,and neglected parasitic diseases was compiled quite successfully with well-done statistical analyzes. I really found the article orginal. I just wanted a few minor corrections and an addition to the discussion on the PDF document. The manuscript could be accepted after they the corrections are made. I've made the corrections on the PDF document that I've loaded.

Reviewer #2: The manuscript entitled “The notable global heterogeneity in the distribution of COVID-19 cases and the association with pre-existing parasitic diseases” by Chang et al. reports a country-level ecological study that assessed the relationship between COVID-19 and parasitic diseases as well as socioeconomic and geographical features. They observed significantly lower COVID-19 incidence rates among malaria-endemic regions, even after potential confounding factors were adjusted. The study was performed with appropriate methods and the results support the authors’ conclusions. Moreover, the manuscript is well written and has good tables and figures. However, the authors should consider performing some changes in the manuscript. The authors provide some plausible pathophysiological hypotheses in order to explain the phenomenon observed in their study. However, in the line 115, they state that the “cytokine storm” observed in some COVID-19 patients is due to excessive Th1 responses. I suggest the reformulation of that text excerpt, since various studies have shown that the Th17-related cytokines seem to be play pivotal roles in the “cytokine storm” observed in the severe forms of the SARS-CoV-2 infection (Wu D, Yang XO. TH17 responses in cytokine storm of COVID-19: An emerging target of JAK2 inhibitor Fedratinib. J Microbiol Immunol Infect. 2020;53(3):368-370; Vatsalya V, Li F, Frimodig JC, et al. Therapeutic Prospects for Th-17 Cell Immune Storm Syndrome and Neurological Symptoms in COVID-19: Thiamine Efficacy and Safety, In-vitro Evidence and Pharmacokinetic Profile. Preprint. medRxiv). Please, write the meaning of the following abbreviations when they appear for the first time in the manuscript: GDP (line 36); SSA (line 53); LMICs (line 66).

Reviewer #3: The study aims to estimate the possible associations of parasitic diseases with Covid-19 incidence in different countries. For this, the authors have used statistical methods considering demographic, socioeconomic, and geographic confounders. They have come up to their conclusions that lower COVID-19 incidence rates were observed in malaria-endemic countries, after accounting for GDP per capita, population above 65 and duration of Covid-19 illness, and that the other parasitic diseases were not significantly associated with the spread of the pandemic. The major problem with the analysis is that they have not accounted for the biases in Covid-19 estimates which vary by countries. The Covid-19 incidence estimates are higher in countries capable of more testing, and vice-versa. The authors themselves mention in the discussion the problem of misclassification of Covid-19 cases due to testing capacity and surveillance quality and that this varies between countries, and that they might not have adjusted for all variables in this analysis. These are real issues in any analysis regarding Covid-19 cases and their associations with other factors. In addition, there are no adequate registries for reporting deaths, cause of death, as well as excess deaths in most low income countries, which again influence the analysis. Therefore, in my opinion, the major confounders are the biases in estimates due to variable testing capacity, true Covid-19 disease burden and excess death estimates, which the authors have not attempted to adjust in their analysis. Unless these various biases in the calculations of the country wise estimates are adjusted in the first place, any analysis on the association of these calculations with other factors may remain biased. At least some estimates for these factors are available, i.e., country wise tests per million population and WHO's excess death estimates attributed to Covid-19 for different countries. The analysis must be attempted after adjusting these confounding factors before this work can be considered.

PLOS authors have the option to publish the peer review history of their article (what does this mean?). If published, this will include your full peer review and any attached files.

Reviewer #1: Yes: Ozlem Ulusan Bagci

Reviewer #2: No

Reviewer #3: Yes: Dr Abhishek Mewara
---

## [Decision Letter · Decision Letter 1]

16 Sep 2022

Dear Prof. Cho,

We are pleased to inform you that your manuscript 'The notable global heterogeneity in the distribution of COVID-19 cases and the association with pre-existing parasitic diseases' has been provisionally accepted for publication in PLOS Neglected Tropical Diseases.

Best regards,

Alberto Novaes Ramos Jr

Academic Editor

Liesl Zuhlke

Section Editor

<style type="text/css">p.p1 {margin: 0.0px 0.0px 0.0px 0.0px; line-height: 16.0px; font: 14.0px Arial; color: #323333; -webkit-text-stroke: #323333}span.s1 {font-kerning: none

</style>

Reviewer's Responses to Questions

**Key Review Criteria Required for Acceptance?**

**Methods**

-Are the objectives of the study clearly articulated with a clear testable hypothesis stated?

-Is the study design appropriate to address the stated objectives?

-Is the population clearly described and appropriate for the hypothesis being tested?

-Is the sample size sufficient to ensure adequate power to address the hypothesis being tested?

-Were correct statistical analysis used to support conclusions?

-Are there concerns about ethical or regulatory requirements being met?

Reviewer #1: Yes.

**Results**

-Does the analysis presented match the analysis plan?

-Are the results clearly and completely presented?

-Are the figures (Tables, Images) of sufficient quality for clarity?

Reviewer #1: Yes.

**Conclusions**

-Are the conclusions supported by the data presented?

-Are the limitations of analysis clearly described?

-Do the authors discuss how these data can be helpful to advance our understanding of the topic under study?

-Is public health relevance addressed?

Reviewer #1: Yes.

**Editorial and Data Presentation Modifications?**

Reviewer #1: Thanks to authors for making the corrections I wanted. I'have made just a few corrections on the manuscript. I've loaded the PDF document. After corrections are made, the manuscript could be accepted without my view.

**Summary and General Comments**

Reviewer #1: Thanks to authors for making the corrections I wanted. I'have made just a few corrections on the manuscript. I've loaded the PDF document. After corrections are made, the manuscript could be accepted without my view.

PLOS authors have the option to publish the peer review history of their article (what does this mean?). If published, this will include your full peer review and any attached files.

Reviewer #1: **Yes: **Ozlem Ulusan Bagci

---

## [Editor Report · Acceptance letter]

3 Oct 2022

Dear Prof. Cho,

We are delighted to inform you that your manuscript, "The notable global heterogeneity in the distribution of COVID-19 cases and the association with pre-existing parasitic diseases," has been formally accepted for publication in PLOS Neglected Tropical Diseases.

Best regards,

Shaden Kamhawi

co-Editor-in-Chief

Paul Brindley

co-Editor-in-Chief
